# Gene delivery of AGAT and GAMT boosts creatine levels in creatine transporter deficiency patient fibroblasts

Chloe Wells[1], Jon Sorgenfrei[1], Sadie L. Johnson[1], Devin Albertson[1], Jared Rutter[2,3,4], Steven Andrew Baker[1]*

1 Department of Pathology, University of Utah, Salt Lake City, Utah, United States of America,
2 Department of Biochemistry, University of Utah, Salt Lake City, Utah, United States of America,
3 Howard Hughes Medical Institute, Salt Lake City, Utah, United States of America, 4 Diabetes & Metabolism Research Center, University of Utah, Salt Lake City, Utah, United States of America

* steven.baker@path.utah.edu

## Abstract

Creatine is a critical metabolite used to buffer cellular energy demands in highly energetic tissues such as the brain and muscle. Genetic defects in endogenous creatine synthesis or transport across cellular membranes lead to a common set of phenotypes referred to as Cerebral Creatine Deficiency Syndrome (CCDS). The most common form of CCDS is Creatine Transporter 1 (CT1) Deficiency (CTD). It accounts for ~70% of cases and results from loss-of-function mutations in the X-linked gene *SLC6A8*. Affected individuals suffer from intellectual disability, autistic-like behaviors, and epilepsy. There are currently no effective therapies for this disorder, but gene therapy has emerged as a potential approach. The two enzymes which comprise the endogenous creatine synthetic pathway (AGAT and GAMT) are selectively expressed by specific cell types throughout the body. However, after synthesized, creatine uptake relies on the protein product of *SLC6A8*, CT1, to transport creatine into target cell types. We hypothesized that gene delivery of *GATM* (encoding AGAT) and *GAMT* into end-user cell types would bypass the need for CT1, allowing for intracellular synthesis of creatine. We tested this strategy in two human cell types: HEK293T cells and primary fibroblasts. Co-delivery of *GATM* and *GAMT* increased internal creatine concentrations by 7.6-fold in HEK293T cells and 12.3-fold in healthy control fibroblasts. We then employed this approach to primary fibroblasts from patients with CTD. This resulted in an up to 11.6-fold increase in intracellular creatine concentrations, far exceeding the intracellular concentration of creatine in healthy control fibroblasts. Importantly, overexpression of AGAT and GAMT resulted in proper targeting of these enzymes to their natural cellular compartment and did not impair the growth of patient fibroblasts. These findings establish gene therapy with *GATM* and *GAMT* as a potential strategy for patients with CTD.

**Data availability statement:** All relevant data are within the manuscript and its Supporting Information files.

**Funding:** CW, JS, and SAB were supported by a grant from the Association for Creatine Deficiencies. SAB was supported by a grant from the Uplifting Athletes' Foundation. SLJ, DA, and SAB were supported by funds from the University of Utah Department of Pathology. JR is an Investigator of the Howard Hughes Medical Institute.

**Competing interests:** I have read the journal's policy and the authors of this manuscript have the following competing interests: JR is a founder of Vettore Biosciences and a member of its scientific advisory board. All other authors declare no competing interests.

## Introduction

Creatine is a metabolite which serves to buffer cellular energy demands by undergoing reversible phosphorylation [1]. Creatine kinases utilize ATP to phosphorylate creatine in periods of energy abundance, for later transfer of a phosphate back to ADP when cellular energetic demands increase [2]. About half of the body's daily creatine requirement is absorbed from food, while the other half is synthesized endogenously, in individuals consuming a Western diet [3]. Endogenous creatine is generated by two enzymes: glycine amidinotransferase, mitochondrial (GATM; also known as L-Arginine:glycine amidinotransferase or AGAT, EC 2.1.4.1), and guanidinoacetate N-methyltransferase (GAMT, EC 2.1.1.2). AGAT transfers the amidino group of arginine onto the amine of glycine to generate the intermediate guanidinoacetate (GAA). GAA is then methylated by GAMT on the same nitrogen, using S-adenosyl methionine (SAM) as a methyl donor, to produce creatine [4]. Loss-of-function mutations in the genes encoding either of these enzymes (*GATM* or *GAMT*, respectively) lead to a disorder referred to as Cerebral Creatine Deficiency Syndrome (CCDS) [5]. Affected individuals exhibit intellectual disability, developmental delay, autistic-like behaviors, and seizures [6]. Defects in *GATM* cause AGAT Deficiency, accounting for less than 10% of CCDS cases, whereas mutations in *GAMT* cause GAMT Deficiency, making up ~20% of individuals with this disease [5]. *GATM* and *GAMT* are located on autosomes and, correspondingly, both disorders exhibit autosomal recessive inheritance.

By contrast, the most common form of Creatine Transporter 1 (CT1) Deficiency (CTD) is caused by loss-of-function mutations in the X-linked gene *SLC6A8* [7]. This gene encodes the principal transporter mediating cellular import of creatine (CT1). These patients comprise approximately 70% of CCDS cases [5]. Affected males typically have a more severe phenotype than females, owing to the presence of a second copy of *SLC6A8* in females, which can compensate for heterozygous mutations depending on X-chromosome inactivation [8].

If diagnosed early, patients with AGAT or GAMT deficiency can be supplemented with creatine in their diets, markedly improving symptoms [9]. For individuals with CTD, creatine supplementation generally does not improve symptoms in the absence of residual transporter function [10]. At present, there are no highly effective therapies for patients with this frequent form of CCDS, but gene therapy via *SLC6A8* delivery is an emerging possibility [11].

Canonically, AGAT activity is thought to be the highest in the kidney [12,13], while GAMT expression is highest in the liver [14]. However, previous characterization of the creatine synthetic pathway by our group [15], and others [16,17], has detected expression in several other organs and tissues. Oligodendrocytes in the brain express both enzymes at high levels [15], suggesting that certain cell types in the body are capable of synthesizing creatine from its amino acid precursors directly. We hypothesized that co-expressing AGAT and GAMT would permit intracellular creatine synthesis in other cell types. Creatine is a zwitterion at physiologic pH [4,18]. Internal creatine synthesis may allow cells to retain this metabolite due to its charge, if membrane transporters for creatine are not expressed. Thus, internal synthesis of creatine has the potential to bypass the need for CT1.

To test this hypothesis, we overexpressed both AGAT and GAMT in HEK293T cells. This resulted in a dramatic increase in intracellular creatine content without salient effects on cell viability. We then used these cells to produce lentiviral particles capable of delivering human *GATM* and *GAMT* to other cell types. Upon delivery to healthy control and CTD patient derived fibroblasts, AGAT and GAMT expression was markedly increased, and both enzymes localized appropriately to their natural compartments inside the cell. Genetic delivery of these enzymes dramatically increased cellular creatine content, indicative of intracellular synthesis. Cells retained AGAT and GAMT expression after several passages without selection, and cellular growth was not impaired in patient fibroblasts with loss-of-function of CT1. These results suggest that boosting creatine synthesis does not have major deleterious effects on cellular physiology and establish a first step proof-of-concept for this strategy as a potential gene therapy for patients with CTD.

## Materials and methods

### Cell lines and culture conditions

HEK293T cells were originally obtained from ATCC (Cat. #CRL-3216) and then gifted by the Rutter laboratory at passage 11 (P11). For primary fibroblasts, the following cell lines were obtained from the NIGMS Human Genetic Cell Repository at the Coriell Institute for Medical Research: Wild-type (WT) control (Cat. #GM28214) skin fibroblasts from a 41-year-old male (passage 2); CT1 mutant $SLC6A8^{\Delta ex10-11/y}$ (Ca. #GM27448) skin fibroblasts from a 4-year-old male (passage 3); and CT1 mutant $SLC6A8^{W556X/y}$ (Ca. #GM27865) skin fibroblasts from a 13-year-old male (passage 2). Cells were grown in a humidified incubator at 37°C and 5% $CO_2$ under atmospheric oxygen. HEK293T cells were cultured in Dulbecco's Modified Eagle Medium (DMEM, Corning, Cat. #10–013-CM) with 10% Fetal Bovine Serum (FBS, Peak Serum, Cat. #PS-FB2), and 1% Penicillin/Streptomycin (Gibco, Cat. #15140–122) in uncoated 10 cm culture plates (Greiner Bio-One, Cat. #664160-TRI) and passaged every three to four days at ~85–95% confluence. Similar conditions were maintained for primary fibroblasts except DMEM was supplemented to 15% FBS and cells were passaged at 80–90% confluence in intervals which varied by cell line, but typically three to seven days with media changes every three to four days.

### Cloning and primers

pLenti CMV Blast empty (w263-1) was a gift from Dr. Alex Bott in the Rutter laboratory and was originally obtained from Addgene (Cat. #17486). This plasmid was modified by site-directed mutagenesis (QuikChange XL Kit, Agilent, Cat. #200516) per the manufacturer's instructions to eliminate the ORF in the 5' gateway arm and insert additional restriction enzyme sites into the multiple cloning site using two primers: pLenti2-Blast-QC.F1 5'-AGAGCTCGTTTAGTGAACCGTACCGGTTTAATTAAACTAGTGGCCGGCCGTCGACTGGATCCGGTACCGA-3' and pLenti2-Blast-QC.R1 5'-AGAGCTCGTTTAGTGAACCGTACCGGTTTAATTAAACTAGTGGCCGGCCGTCGACTGGATCCGGTACCGA-3'. The resulting plasmid was verified by Sanger sequencing, renamed pLenti2-Blast and used for lentiviral production in this study. *Lentivirus Production*. HEK293T cells were plated in a 10 cm dish in DMEM, 10% FBS, and 1% Penicillin/Streptomycin. The following day, two hours prior to transfection, the media was changed to fresh media. HEK293T cells were then transfected with 4 μg psPAX2, 1 μg pVSV-G, and 3 μg of cargo plasmid (pLenti2-Blast) DNA using Lipofectamine 2000 (Thermo Fisher, Cat. #52887) per the manufacturer's instructions. The next morning, media was exchanged to remove the transfection mix. For three collections, 8–16 hours apart, the media was exchanged for fresh media and the post-incubation media was collected, pooled, and stored at 4°C. After three collections, the pooled media was centrifuged for 5 minutes at 500xg at 4°C to pellet any transferred cells. The supernatant was then filtered through a 0.22-μm Steriflip filter unit (Millipore, Cat. #SCGP00525). Filtered virus was divided into 1 or 2 mL aliquots and stored at -80°C until needed. Viral aliquots of each construct were titered with HEK293T cells in a 3-dilution series scoring for blasticidin resistance (Gibco, Cat. #A11139-03 used at 10 μg/mL final).

## Western blots

HEK293T cells (between P12 and P20) were plated at a ratio of 1:5 in a 6-well plate in DMEM, 10% FBS, and 1% Penicillin/Streptomycin. The following day, cells were transfected with 1 μg of DNA using Lipofectamine 2000 per the manufacturer's protocol. After 36–48 hours, the media was aspirated and cells were harvested by pipetting with 1 mL ice cold 1xPBS (Gibco, Cat. #10010–023) and centrifuged for one minute at 16,000xg to pellet. The PBS was aspirated and cells were lysed in 500 μL Extraction Buffer (100 mM Tris-Cl pH 7.4, 2% SDS) for 15 minutes, rotating at room temperature. The lysates were then centrifuged at room temperature for ten minutes at 16,000xg. The gelatinous DNA pellet was removed and 30 μL of the supernatant was added to 30 μL of 2x Laemmli sample buffer (Thermo Fisher, Cat. #NP0007). Samples were either used immediately or frozen for later at -80°C. For SDS-PAGE, samples were incubated at 95°C for five to ten minutes, then loaded into 10-well, or 15-well 4–15% precast gels (Bio-Rad Cat. #4561084, Cat. #4561086). Samples were electrophoresed until completion and then transferred to 0.45 μm nitrocellulose membranes (Amersham Protran, Cat. # 10600002) for approximately one hour. Blots were blocked with 5% BSA in 1xTBST for 30 minutes, prior to incubation with primary antibody overnight in blocking buffer. Primary antibodies included: mouse anti-AGAT (Novus, Cat. #NBP2–00984, 1:300); rabbit anti-GAMT (Biorbyt, Cat. #orb247514, 1:300); chicken anti-GAPDH (Abcam, Cat. #ab83956); rabbit anti-Histone H3 (Sigma Aldrich, Cat. #07–690); and mouse anti-V5 (BioRad, Cat. #MCA1360). The following day, membranes were washed x4 with 1xTBST prior to incubation with fluorescent secondary antibodies in 5% skim milk in 1xTBST for one hour at room temperature. Secondary antibodies used were: Donkey anti-mouse 790 (Jackson ImmunoResearch, Cat. #715-655-151); Donkey anti-rabbit 680 (Jackson ImmunoResearch, Cat. #711-625-152); and Donkey anti-chicken 488 (Jackson ImmunoResearch, Cat. #703-545-155). After four washes, membranes were imaged with a LI-COR Odyssey CLx or BioRad ChemiDoc MP. The same procedure was followed for human fibroblasts except that after aspirating the culture media, the cells were incubated with ice cold PBS containing 0.5 mM EDTA for five to ten minutes to facilitate collection by pipetting.

## Creatine assay and protein quantification

For HEK293T experiments, cells were plated in a 6-well plate as described for western blotting. After 36–48 hours post-transfection, the media was changed to creatine-free media (DMEM Corning, Cat. #10–013-CM, with 10% dialyzed FBS Gibco, Cat. #A33820-01, and 1% Penicillin/Streptomycin Gibco, Cat. #15140–122) supplemented with L-arginine, glycine, and methionine to 1 mM above the standard formula for DMEM. Approximately 24 hours later, the media was aspirated and cells were gently washed twice with ice cold PBS. Cells were collected by pipetting with a third PBS wash and centrifuged at 16,000xg for one minute to pellet. PBS was aspirated and the cell pellet was resuspended in 1 mL of -80°C 80% methanol. Cell lysates were rotated for 15 minutes at 4°C before spinning down in a centrifuge at 16,000xg for 10 minutes at 4°C. All supernatant was collected and placed in a vacuum centrifuge overnight at room temperature to allow the methanol to evaporate, leaving the dried metabolite pellet. The insoluble protein pellet that was left in the tube after methanol extraction was resuspended in 1 mL Extraction Buffer, rotating for 15 minutes at room temperature. Once homogenous, the protein suspension was centrifuged for ten minutes at room temperature at 16,000xg to remove insoluble material. A 25 μL sample of the supernatant was used to measure the protein concentration relative to a 25 μL BSA standard series with the Pierce BCA Protein Assay (Thermo Scientific, Cat. #23225) per the manufacturer's instructions.

The next morning, the dried metabolite pellet was taken from the vacuum centrifuge and resuspended in a known amount (either 200 or 310 μL) of Creatine Assay Buffer (CAB, Sigma Aldrich, Cat. #MAK079A). The resuspended pellet was allowed to equilibrate by rotating for 15 minutes at room temperature before either being used immediately or frozen at -20°C for later analysis. Cytoplasmic creatine was measured from this sample using a commercially available Creatine Assay Kit (Sigma Aldrich, Cat. #MAK079) on a BioTek Synergy Neo2 plate reader. The plate reader was programmed to measure in kinetic mode once every 30 seconds for 30 minutes. Measurements were standardized to a known creatine standard series diluted in CAB per the manufacturer's instruction. The slope of 570 nm absorbance produced by a 50 μL

aliquot of the sample was averaged in duplicate and compared to the slope of the creatine standard dilution series. The resulting concentration was multiplied by the volume of CAB used to resuspend the total metabolite pellet to produce the measurement for total cellular creatine in each cell pellet. These values were normalized to the total protein mass measured by the BCA assay to produce values in nmoles creatine per mg total protein.

For WT and CTD patient fibroblasts, stable empty vector or AGAT and GAMT expressing lines were established by infecting the parental lines with lentiviral particles at similar titer. After selection for three days with blasticidin, these lines were passaged in standard media, (DMEM with 15% FBS and 1% Penicillin/Streptomycin), until the start of experiments. For the creatine assay, stable lines were plated at~10–30% confluence in a 15 cm culture plate and allowed to grow for one month, exchanging the standard media every three to five days. This long culture process was found to be necessary to produce enough cellular material for creatine measurements that were in the linear range for our Creatine Assay Kit. Empty vector controls and AGAT and GAMT co-expressing lines were produced, fed, and treated under identical conditions for each parental line (WT, $SLC6A8^{\Delta ex10-11/y}$, and $SLC6A8^{W556X/y}$). After 1 month of growth, the plates were completely confluent. The media was exchanged for creatine-free media (DMEM with 15% dialyzed FBS, and 1% Penicillin/Streptomycin) and the cells were allowed to incubate for three days. After three days, a sample of the media was collected and frozen at -20°C for direct measurement with the Creatine Assay Kit. The remaining media was aspirated and the cells were washed twice with PBS. A third wash was then added containing PBS with 0.5 mM EDTA and the cells were allowed to incubate at room temperature for approximately five minutes to loosen from the dish. The cells were then collected by pipetting and pelleted at room temperature at 3,220xg for one minute in a 15 mL conical tube. The third wash was aspirated, leaving a small meniscus, and then a fourth wash with 1 mL of PBS was used to resuspend the pellet and transfer it to a 1.5 mL Eppendorf tube. This was then centrifuged at 4°C and 16,000xg for 1 minute to pellet the cells into a tight pellet prior to aspirating all of the PBS. This pellet was then used for metabolite extraction with 1 mL of -80°C 80% methanol, identical to the extraction for HEK293T cells described above. The remaining steps for metabolite extraction, drying, resuspension, and quantification were identical to those used for HEK293T cells, as was the quantification of total protein mass using the BCA assay.

*Immunofluorescence*. Sterile coverslips (FisherBrand, 18 mm circle #1, Cat. #12541005) were placed in a 12-well culture plate (Greiner Bio-One, Cat. #665180). These were coated with poly-D-lysine (EMD Biosciences, Cat. #A-003-E) 100 µg/mL in PBS at room temperature for 2 hours, air dried, and rinsed x3 with PBS prior to plating fibroblasts at~10–30% confluence in standard media. After ~2–5 days, when cells reached 30–70% confluence, they were rinsed with PBS once and then immediately fixed with 4% PFA in PBS. After 15 minutes, the fix was removed, the cells were washed x3 in PBS, and then permeabilized with PBS-0.1% Triton X-100 for 15 minutes. After permeabilization the cells were incubated for one hour in blocking solution (5% Normal Goat Serum+1% BSA in PBS 0.05% Tween-20). The block was removed and then coverslips were incubated overnight at 4°C with primary antibody in blocking solution. Primary antibodies were mouse anti-AGAT (Novus, Cat. #NBP2–00984, 1:300) and rabbit anti-GAMT (Biorbyt, Cat. #orb247514, 1:300). The next day, coverslips were washed x3 with PBS and then incubated for one hour at room temperature with secondary antibody in blocking solution. Secondary antibodies were: Polyclonal goat anti-mouse 488 (Thermo, Cat. #A11001, 1:1000); and Polyclonal goat anti-rabbit 594 (Thermo, Cat. #A11012, 1:1000). Coverslips were then washed once in PBS with 5 µg/mL DAPI for 15–30 minutes, then washed an additional three times in PBS before mounting on SuperFrost Plus Slides (VWR, Cat. #48311–703) in Vectashield Vibrance Antifade Mounting Media (Vector Labs, Cat. #H-1700). Slides were imaged on a Zeiss LSM880 Confocal Microscope and images were processed with ImageJ.

## Growth assay

Fibroblasts from each line were trypsinized, and following neutralization, the cell suspension was counted using a manual hemocytometer (Fisher Scientific, Cat. #0267151B). Cells were diluted to 50,000 cells per 3 mL of either standard media (DMEM+15% FBS+P/S) or standard media supplemented with creatine precursors (L-arginine, glycine, and methionine

at 1 mM final above the concentration in standard media). Three mL of the cell suspension was plated in each well of a 6-well plate (Costar, Cat. #3516) and allowed to settle in a tissue culture incubator for one to four hours. The plates were then placed in an IncuCyte S3 Live-Cell Analysis System (Sartorius) and imaged using the phase channel through a 10x objective (16 images per well, every 6 hours for 72 hours total). Images were then processed using the Analysis Wizard within IncuCyte's manufacturer provided software, computing the raw percentage confluence averaged for all 16 images from each well at each time point. The raw confluence values were normalized to the maximum confluence achieved at the end of three days for the highest confluence condition and plotted as a percentage of the maximum confluence for all conditions collected in the same batch. Each batch contained three replicates of empty vector control and three replicates of AGAT and GAMT overexpressing stable fibroblast lines from the same parental genotype.

### Statistical analysis

Data were processed and analyzed using R 4.2.2. For t-tests, the *t.test()* function from base R was used with the arguments *alternative = "two.sided"* and *var.equal = T*. For one-way ANOVA the *aov()* function from base R was used with default settings. For the exact two-sample Kolmogorov-Smirnov (K-S) test the *ks.test()* function from base R was used with *alternative = "two.sided"*. A *P-value* of $P < 0.05$ was used for statistical significance.

### Results

We constructed lentiviral plasmids to express WT full-length human AGAT or GAMT either untagged or tagged with a short peptide (V5) driven by the ubiquitously expressed CMV promoter. In order to assess for proper expression of these constructs, we transfected them into HEK293T cells in isolation or various combinations (Fig 1A and 1B). After 36–48 hours the cells were collected, lysed, and analyzed by western blotting. Transfection with an empty vector control revealed that HEK293T express virtually no AGAT at the protein level but have a small amount of endogenous GAMT expression (Fig 1A). Transfection with human *GATM* or *GAMT* dramatically increased the expression of each protein. When transfected together, both proteins were highly co-expressed, with minimal impact on the abundance of each other, compared to singly transfected cells (Fig 1A). To compare the relative abundance of these two proteins, both were transfected with the same V5 tag and analyzed by western blotting (Fig 1B). This revealed similar abundance of both proteins and unchanged abundance of AGAT-V5 co-expressed with GAMT-V5, when compared to AGAT-V5 transfected only cells (Fig 1B). Importantly, the cells from each well were lysed in an equal volume of buffer and loaded onto the SDS-PAGE gel using an identical sample volume, revealing uniform abundance of loading controls (GAPDH and Histone H3, Fig 1A). These data indicate that transient overexpression of AGAT and GAMT over this time period has minimal impacts on cell abundance for HEK293T cells.

We next performed a similar experiment, but instead of lysing cells 36–48 hours post-transfection, the media was exchanged for creatine-free media supplemented with creatine precursor amino acids (L-arginine, glycine, and methionine, 1 mM each). Twenty-four hours later, the cells were washed and intracellular metabolites were extracted and analyzed for creatine content (Fig 1C). The abundance of creatine was markedly increased in cells co-expressing AGAT and GAMT relative to empty vector controls (208.0 vs 27.5 nmoles/mg protein, respectively, or 7.6-fold, $P < 0.05$, two-sided t-test). The same result was obtained in cells co-expressing AGAT-V5 and GAMT-V5 (238.1 nmoles/mg protein, or 8.7-fold increased relative to control, $P < 0.05$, two-sided t-test). Although the effect was smaller when AGAT-V5 was expressed alone, it also reached statistical significance (54.0 nmoles/mg protein, or 2.0-fold increased relative to control, $P < 0.05$, two-sided t-test). The reduced amount of creatine synthesized by AGAT-V5 overexpression alone likely reflects the low, but detectable, endogenous expression of GAMT in HEK293T cells (see Fig 1A). To further control for cellular abundance/survival upon transient overexpression of AGAT and GAMT, we quantified the total cellular protein from each well (Fig 1D). This revealed no significant difference in total protein abundance between conditions (one-way ANOVA, $F(5,18)$).

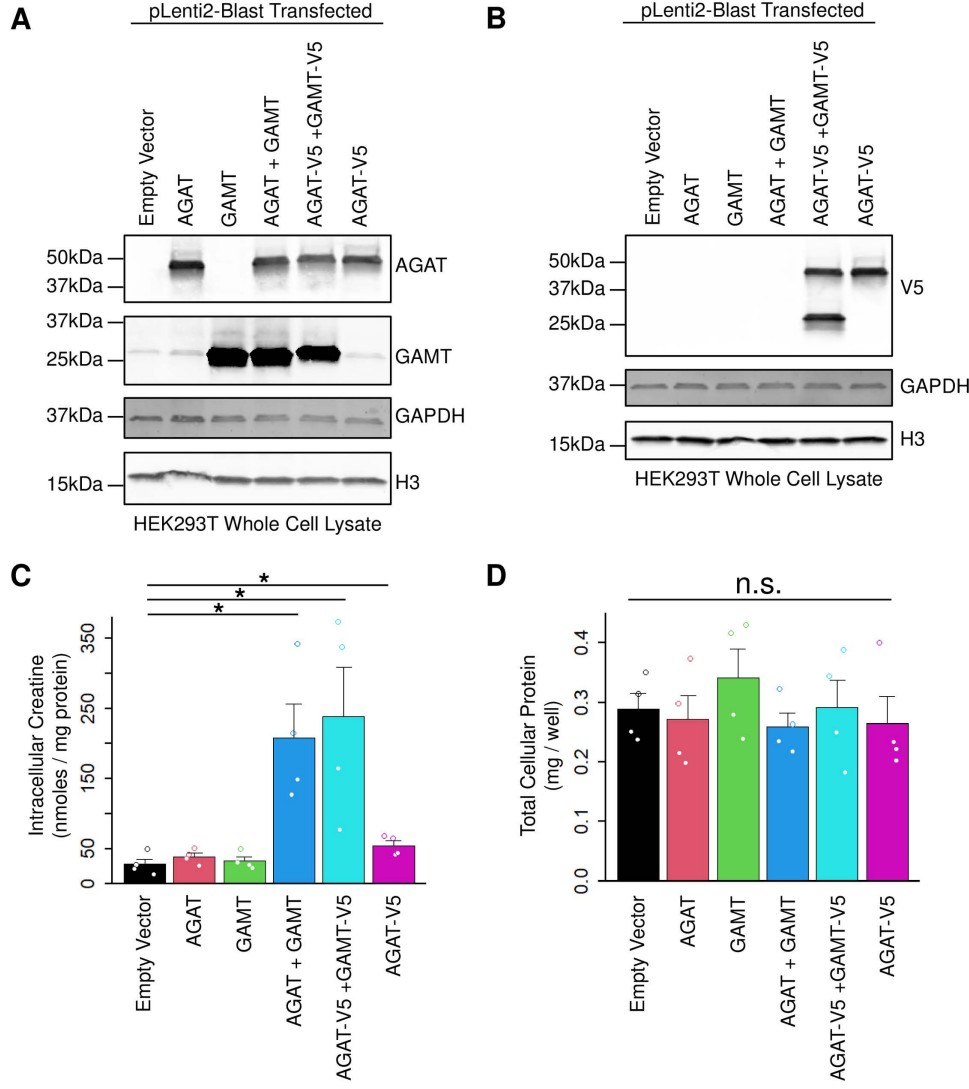

**Fig 1. Expression of AGAT and GAMT in HEK293T cells.** (A) HEK293T cells were transfected with the indicated plasmid. Cells were then lysed and analyzed by western blotting with the indicated antibody. N = 3. (B) Same lysates as in (A) were probed for the V5 tagged form of AGAT and GAMT by western blot. N = 3. (C) Cells transfected as in (A) were cultured in creatine-free media supplemented with amino acid precursors for 24 hours. Cells were lysed and the total cellular creatine was quantified. N = 4. Two-sided t-test * indicates $P < 0.05$. (D) Same experiment as in (C) plotting the total cellular protein measured. N = 4. n.s. indicates not significant for one-way ANOVA. Error bars show SEM.

Having established that our constructs were functional, we then used HEK293T cells to prepare lentiviral particles for infection of other cell types. We started by infecting human WT control fibroblasts and assessing expression at the protein level (Fig 2A and **2B**). Delivery of human *GATM* or *GAMT* to these cells resulted in abundant overexpression of these proteins relative to an essentially undetectable background expression level (Fig 2A). Cells were selected with blastici-din 24 hours post-infection, to eliminate uninfected cells, and then allowed to grow for two to three days (in blasticidin) prior to harvesting. All the surviving cells from each well were lysed in an equal amount of buffer and an equal volume of sample was loaded onto SDS-PAGE gels. The total amount of GAPDH detected in each lane was unchanged and the cells appeared healthy at harvest suggesting that overexpression of AGAT and GAMT was not deleterious for cell viability

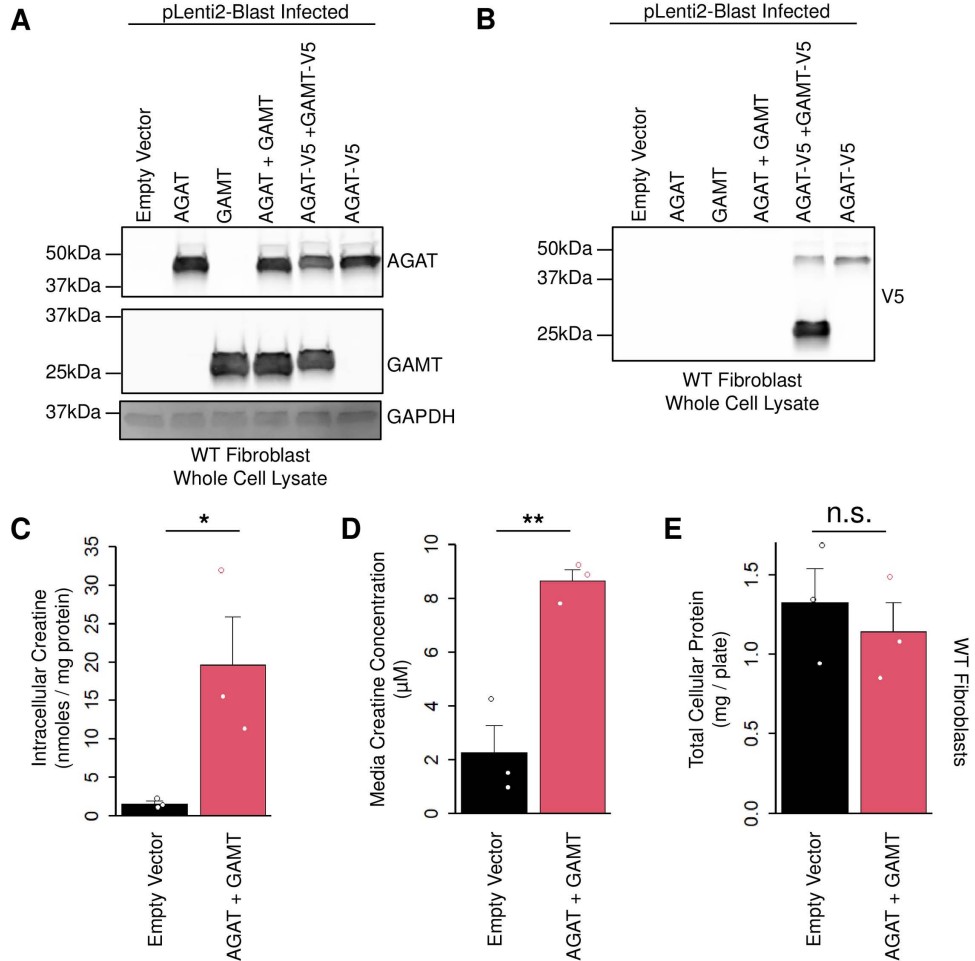

**Fig 2. Total cellular expression of AGAT and GAMT in WT fibroblasts.** (A) WT control fibroblasts were infected with the indicated lentivirus. Cells were then lysed and analyzed by western blotting with the indicated antibody. N = 3. (B) Same lysates as in (A) were probed for the V5 tagged form of AGAT and GAMT by western blot. N = 3. (C) WT fibroblasts stably expressing the indicated transgenes were cultured in creatine-free media supplemented with amino acid precursors for 3 days. Cells were lysed and the total cellular creatine was quantified. N = 3. Two-sided t-test * indicates $P < 0.05$. (D) Same experiment as in (C) plotting the media creatine concentration measured. N = 3. Two-sided t-test ** indicates $P < 0.01$. (E) Same experiment as in (C) plotting the total cellular protein measured. N = 3. Two-sided t-test n.s. indicates not significant. Error bars show SEM.

(Fig 2A). We also compared the abundance of AGAT-V5 and GAMT-V5 by probing for the V5 tag in the same lysates from Fig 2A which revealed a relative overabundance of GAMT-V5 compared to AGAT-V5 (Fig 2B). This likely owes to differences in the subcellular distributions of these two proteins. AGAT is restricted to mitochondria, whereas GAMT distributes throughout the nucleocytoplasm [15]. However, slight differences in viral titer or better cell survival/proliferation of cells with higher GAMT expression relative to AGAT is also possible.

We then compared the abundance of intracellular creatine in WT fibroblasts infected with both *GATM* and *GAMT* relative to cells infected with empty vector. These cells were selected in blasticidin 24 hours post-infection for three days and then allowed to grow without selection for approximately one month (the equivalent of approximately 6–8 passages). After this prolonged growth phase, the media in the dish was exchanged for creatine-free media supplemented with creatine amino acid precursors for three days. When metabolites were extracted from washed cells, the amount of intracellular creatine was found to be markedly increased by co-expression of AGAT and GAMT (Fig 2C, 19.7 vs 1.6 nmoles/mg

protein, or 12.3-fold increased relative to control, $P < 0.05$, two-sided t-test). Prior to harvesting the cells, a sample of the creatine-free media was collected and measured to determine the extracellular creatine content produced (Fig 2D). This revealed an increased concentration of creatine in the media to 8.7 µM relative to 2.3 µM from empty vector controls (3.8-fold, $P < 0.01$, two-sided t-test). Importantly, we did not observe a noticeable difference in the total number of cells after one month of growth in the 15 cm dish used for these experiments, confirmed by quantifying the total cellular protein (Fig 2E, 1.3 vs 1.1 mg in control vs AGAT and GAMT, respectively, not significant, two-sided t-test).

Immunofluorescence for AGAT and GAMT was performed on WT fibroblasts stably expressing these transgenes to confirm expression after several passages without blasticidin selection (Fig 3A and 3B). Empty vector control cells displayed only background signal throughout the cytoplasm when stained for AGAT. In contrast, cytoplasmic and low level nuclear signal was detected when these cells were stained for GAMT (Fig 3A and 3B). In cells stably expressing the *GATM* and *GAMT* transgenes, tubular and punctate signal could be detected for AGAT throughout the cytoplasm, consistent with its known localization to mitochondria [15]. The GAMT signal in these cells was also much more pronounced than empty vector controls and localized throughout the nucleocytoplasm (Fig 3A and 3B). These data indicate that stably infected WT fibroblasts target these enzymes to the appropriate compartment and markedly upregulate their abundance relative to the endogenous level of expression.

We obtained two CTD patient fibroblast lines donated by males with complete loss-of-function mutations in *SLC6A8*. We infected the first mutant line (*SLC6A8$^{\Delta ex10-11/y}$*) with our *GATM* and *GAMT* lentiviral particles and analyzed their expression after blasticidin selection (Fig 4A and 4B). Consistent with our results in WT fibroblasts, delivery of each transgene led to dramatically increased expression of AGAT and GAMT without noticeable impacts on cell abundance as determined by GAPDH loading (Fig 4A). We probed the same lysates for V5 to determine the relative expression of AGAT-V5 and GAMT-V5 in cells co-expressing these transgenes (Fig 4B). Like WT fibroblasts, this revealed an overabundance of GAMT-V5 relative to AGAT-V5 in CT1 mutant fibroblasts.

We then generated CT1 mutant fibroblasts stably expressing AGAT and GAMT from both patients with different loss-of-function mutations (*SLC6A8$^{\Delta ex10-11/y}$* and *SLC6A8$^{W556X/y}$*). We cultured these cells for approximately one month (the equivalent of approximately 6–8 passages) without blasticidin selection and then exchanged their media for creatine-free media supplemented with creatine amino acid precursors to determine their potential to synthesize creatine (Fig 4C-H). After three days, for the first CT1 mutant line (*SLC6A8$^{\Delta ex10-11/y}$*), we observed a 3.5-fold increase in intracellular creatine

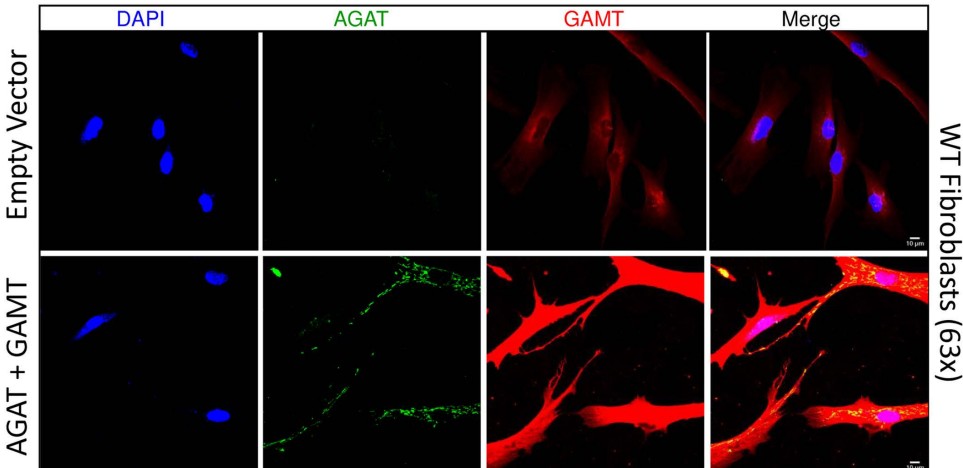

**Fig 3. Localization of delivered AGAT and GAMT in WT fibroblasts.** Immunofluorescence (IF) of WT fibroblasts stably expressing AGAT+GAMT and empty vector infected controls imaged at 63x magnification. N = 4. Scale bars indicate 10 µm.

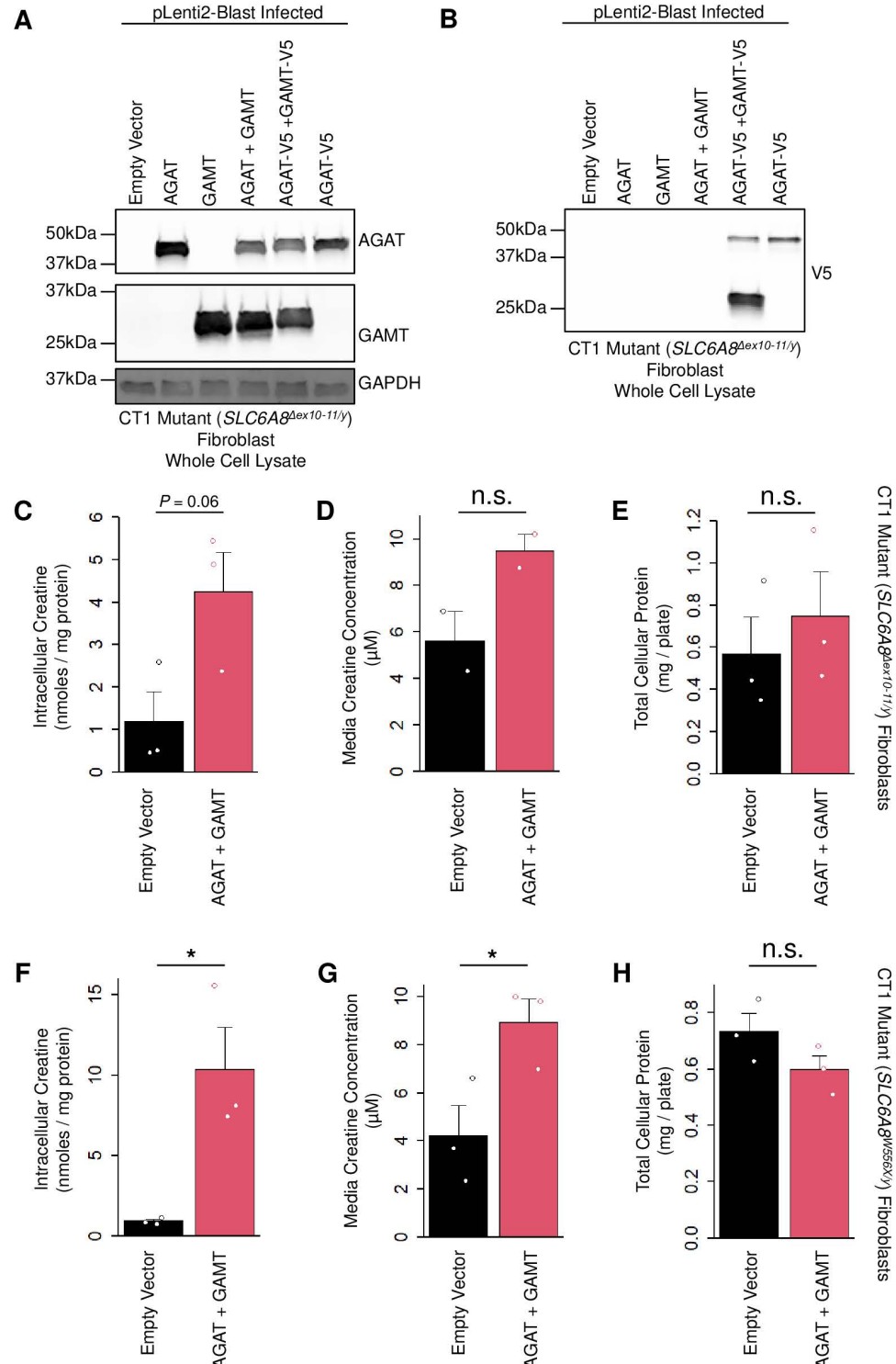

**Fig 4. Total cellular expression of AGAT and GAMT in CT1 mutant fibroblasts.** (A) Fibroblasts from a patient with CT1 loss-of-function ($SLC6A8^{\Delta ex10-11/y}$) were infected with the indicated lentivirus. Cells were then lysed and analyzed by western blotting with the indicated antibody. N = 3. (B) Same lysates as in (A) were probed for the V5 tagged form of AGAT and GAMT by western blot. N = 3. (C) The same CT1 loss-of-function fibroblasts stably expressing the indicated transgenes were cultured in creatine-free media supplemented with amino acid precursors for three days. Cells were

lysed and the total cellular creatine was quantified. N = 3. Two-sided t-test $P = 0.06$. (D) Same experiment as in (C) plotting the media creatine concentration measured. N = 2. Two-sided t-test n.s. indicates not significant. (E) Same experiment as in (C) plotting the total cellular protein measured. N = 3. Two-sided t-test n.s. indicates not significant. Error bars show SEM. (F) A second line of CT1 loss-of-function fibroblasts (*SLC6A8*$^{W556X/y}$) stably expressing the indicated transgenes were cultured in creatine-free media supplemented with amino acid precursors for three days. Cells were lysed and the total cellular creatine was quantified. N = 3. Two-sided t-test * indicates $P < 0.05$. (G) Same experiment as in (F) plotting the media creatine concentration measured. N = 3. Two-sided t-test * indicates $P < 0.05$. (H) Same experiment as in (F) plotting the total cellular protein measured. N = 3. Two-sided t-test n.s. indicates not significant. Error bars show SEM.

although this increase just failed to reach statistical significance (Fig 4C, 4.2 vs 1.2 nmoles/mg protein relative to empty vector controls, $P = 0.061$, two-sided t-test). The amount of creatine in the media collected immediately before harvesting these cells was not significantly different (Fig 4D, 9.5 relative to 5.6 µM from empty vector controls, two-sided t-test). The total cellular protein was also not significantly different (Fig 4E, 0.6 vs 0.7 mg in control vs AGAT and GAMT, respectively, two-sided t-test) indicating that expression of these transgenes did not impact cell abundance. For the second CT1 mutant line (*SLC6A8*$^{W556X/y}$) we observed an 11.6-fold increase in intracellular creatine (Fig 4F, 10.4 vs 0.9 nmoles/mg protein relative to empty vector controls, $P < 0.05$, two-sided t-test). There was also significantly more creatine in the media collected immediately before harvesting these cells (Fig 4G, 8.9 relative to 4.2 µM from empty vector controls, or 2.1-fold increased, $P < 0.05$, two-sided t-test). Importantly, there was again no significant difference in total protein from the harvested cells expressing AGAT and GAMT relative to empty vector controls (Fig 4H, 0.7 vs 0.6 mg, respectively, two-sided t-test). Taken together, these data indicate that co-expression of AGAT and GAMT in CTD patient fibroblasts can increase intracellular creatine up to 11.6-fold.

Next, both CT1 mutant fibroblast lines were analyzed for transgenic protein expression by immunofluorescence (Fig 5 and 6). AGAT expression was undetectable in empty vector controls at low magnification (Fig 5A and 6A). Any signal detected in empty vector controls at high magnification (Fig 5B and 6B) did not localize to mitochondria and is therefore likely background. These data are consistent with essentially no expression of endogenous AGAT in these cells seen by western blot (see Fig 4A). In *GATM* and *GAMT* transgenic CT1 mutant cells, the AGAT signal localized well to tubular/punctate structures consistent with mitochondrial targeting in both *SLC6A8*$^{\Delta ex10-11/y}$ mutants (Fig 5) and *SLC6A8*$^{W556X/y}$ (Fig 6). GAMT expression was detected throughout the nucleocytoplasm in empty vector controls but was dramatically

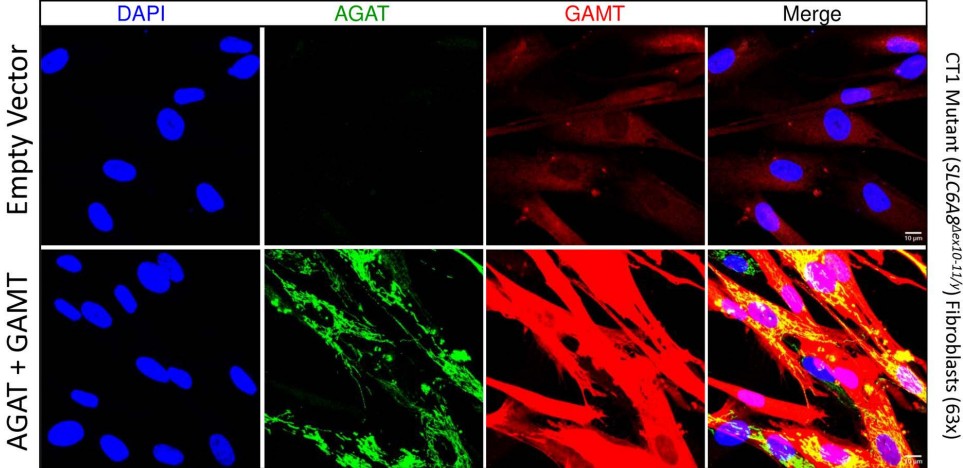

**Fig 5. Localization of delivered AGAT and GAMT in CT1 mutant (*SLC6A8 Δex10-11/y*) fibroblasts.** IF of CT1 loss-of-function patient fibroblasts stably expressing AGAT + GAMT and empty vector infected controls imaged at 63x magnification. N = 4. Scale bars indicate 10 µm.

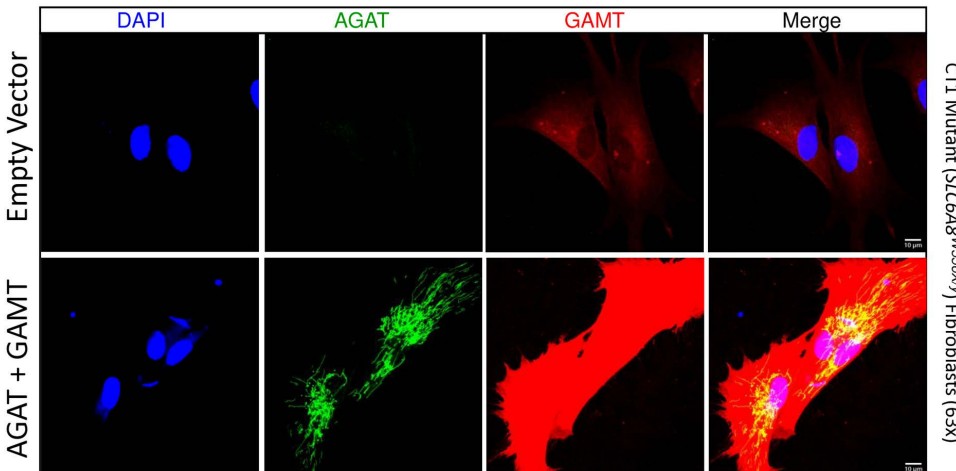

**Fig 6. Localization of delivered AGAT and GAMT in CT1 mutant (*SLC6A8W556X/y*) fibroblasts.** IF of a second line of CT1 loss-of-function patient fibroblasts stably expressing AGAT+GAMT and empty vector infected controls imaged at 63x magnification. N=4. Scale bars indicate 10 μm.

increased in *GATM* and *GAMT* stable transgenic fibroblasts (Fig 5 and 6). These data indicate that AGAT and GAMT target appropriately to their respective subcellular compartment. Furthermore, immunofluorescence analysis was carried out after several passages without blasticidin selection, suggesting that high level AGAT and GAMT overexpression does not cause major deleterious effects on CT1 mutant fibroblast survival.

To support these initial observations, we then performed growth rate quantification on WT and CT1 mutant fibroblasts stably overexpressing AGAT and GAMT (Fig 7). We imaged fibroblasts plated at 50,000 cells per well every six hours over the course of three days in standard media (DMEM+15% FBS+P/S). For CT1 mutant fibroblasts, we also performed these experiments in the same media supplemented with creatine precursor amino acids (Fig 7A). In standard media, WT fibroblast exhibited an indistinguishable growth curve whether they overexpressed AGAT and GAMT or an empty vector control (Fig 7B, not significant, two-sided K-S test). In the same media, the first CT1 mutant fibroblast line (*SLC6A8$^{Δex10-11/y}$*) expressing AGAT and GAMT outgrew empty vector controls (Fig 7C, $P < 0.0001$, two-sided K-S test). When the media was supplemented with creatine precursors, cells overexpressing AGAT and GAMT also outgrew their empty vector controls (Fig 7D, $P < 0.001$, two-sided K-S test). Similarly, the second line of CT1 mutant fibroblasts (*SLC6A8$^{W556X/y}$*) stably overexpressing AGAT and GAMT outgrew their empty vector controls (Fig 7E, $P < 0.05$, two-sided K-S test). This effect was observed again when creatine precursors were supplemented in the media (Fig 7F, $P < 0.0001$, two-sided K-S test). In summary, stable overexpression of AGAT and GAMT does not impair the growth of CTD patient fibroblasts irrespective of precursor supplementation.

## Discussion

Creatine is an essential molecule required for physiologic brain activity. Although patients with CTD have normal functioning copies of *GATM* and *GAMT*, the endogenous expression of these synthetic enzymes cannot compensate for the loss of CT1. This holds true despite the observation that AGAT and GAMT are expressed by various cell types throughout the body [15,17], including oligodendrocytes in the brain. Transport of creatine across the blood-brain barrier is poor when this metabolite is supplemented in the diet [19], evidenced by the slow recovery of brain creatine levels in patients with AGAT or GAMT Deficiency on oral creatine therapy [9]. In those patients (which have biallelic loss of *GAMT* or *GATM*, respectively), replenishment of creatine abundance in the brain requires many months of dietary supplementation. This is despite the fact that creatine supplementation vastly increases plasma creatine levels [20] in patients with AGAT or GAMT

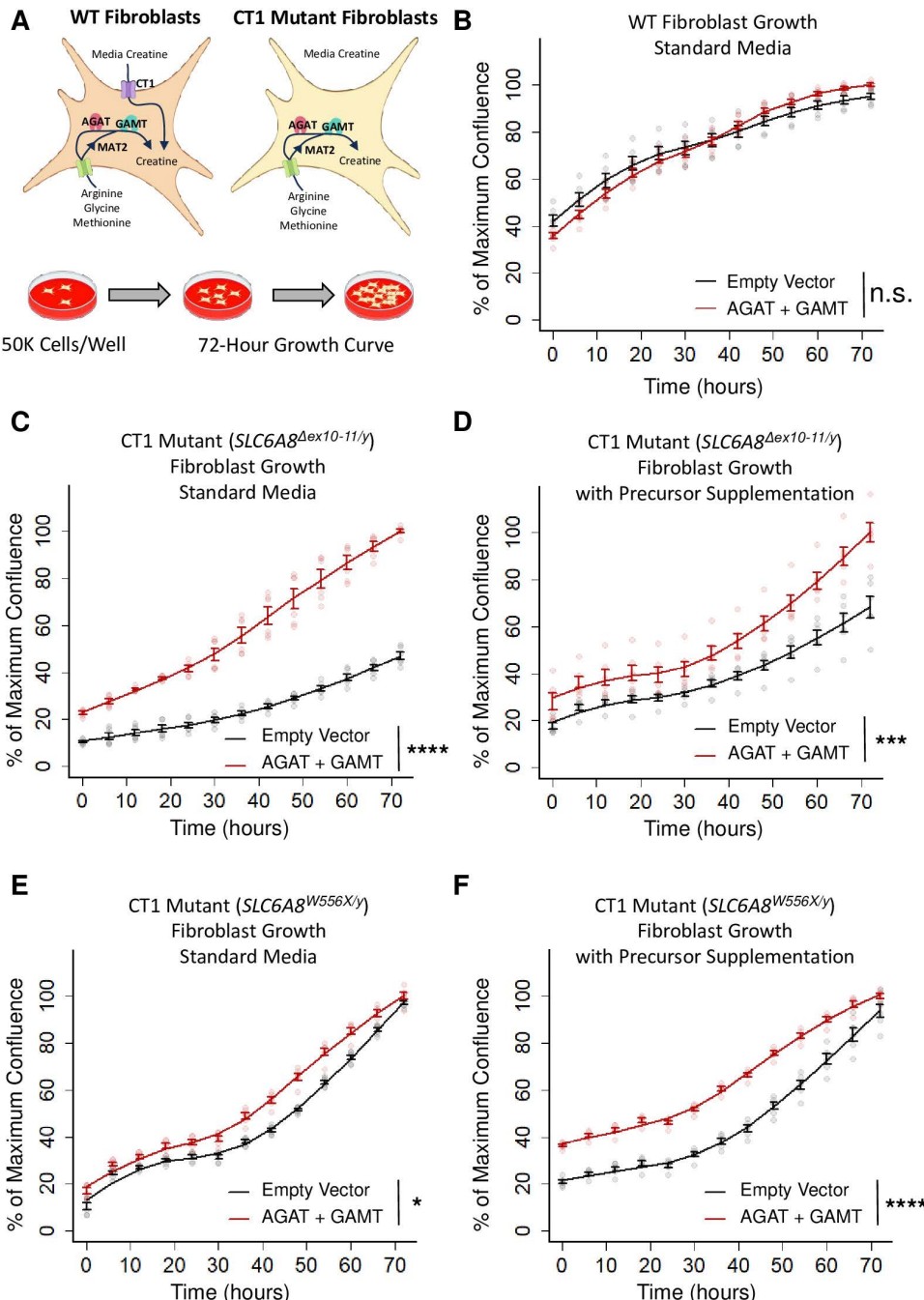

**Fig 7. Growth characteristics of fibroblasts stably expressing AGAT and GAMT.** (A) A model depicting the two sources of intracellular creatine. Creatine is either imported via CT1 or synthesized from its amino acid precursors: L-arginine; glycine; and methionine. The latter is converted into SAM via MAT2. Created with BioRender.com. (B) Growth of WT fibroblasts stably expressing the indicated constructs in standard media (DMEM + 15% FBS + P/S). N = 6. Two-sided K-S test n.s. indicates not significant. (C) Growth of CT1 loss-of-function ($SLC6A8^{\Delta ex10-11/y}$) patient fibroblasts stably expressing the indicated constructs in standard media. N = 6. Two-sided K-S test **** indicates $P < 0.0001$. (D) Similar experiment to (C) except media was supplemented with creatine precursor amino acids to 1 mM above standard media. N = 6. Two-sided K-S test *** indicates $P < 0.001$. (E) Growth of CT1 loss-of-function ($SLC6A8^{W556X/y}$) patient fibroblasts stably expressing the indicated constructs in standard media. N = 6. Two-sided K-S test * indicates $P < 0.05$. (F) Similar experiment to (E) except media was supplemented with creatine precursor amino acids. N = 6. Two-sided K-S test **** indicates $P < 0.0001$. Lines indicate a LOESS curve fit to the mean at each timepoint. Error bars show mean +/- SEM.

Deficiency and both groups express a functional CT1 transporter. If the brain can endogenously produce creatine, and CT1 is not an efficient transporter of creatine across the blood-brain barrier, why are creatine levels decreased by ~80% [21] in the brains of patients with CTD? One hypothesis is that creatine is produced by oligodendrocytes in the brain for local use by end-user cell types, but it cannot be taken up and concentrated in those cells without CT1. In support of this hypothesis, the cerebral spinal fluid (CSF) of patients with CTD contains a normal to even slightly elevated concentration of creatine [22,23].

Creatine is important to support variable energy demands by highly active tissues such as cardiac and skeletal muscle [24]. Importantly, both of these myocyte lineages uniquely express a form of creatine kinase (sarcomeric creatine kinase mitochondrial, ScCKmit) that localizes to mitochondria (encoded by *CKMT2*) [25]. Within the brain, ScCKmit is not expressed. Instead, a related mitochondrial kinase (Ubiquitous creatine kinase mitochondrial, UbCKmit, encoded by *Ckmt1* in mice), is exclusively expressed at high levels by neurons [26]. This feature and the known energetic demands of neuronal membrane polarization and synaptic signaling [27], suggest that neurons are a principal end-user of creatine in the brain. Thus, even though patients with CTD are likely capable of synthesizing creatine within the brain, the net deficiency of this metabolite may result from impaired uptake of creatine from the CSF by neurons without a functional copy of CT1. One means of bypassing this requirement is to permit neurons to synthesize creatine within their own cytoplasm by overexpressing AGAT and GAMT. It is worth noting that many examined neuronal populations express GAMT at low levels but do not express AGAT [15].

Similar to neurons, we determined here that HEK293T cells express low levels of GAMT but do not express detectable levels of AGAT. When AGAT-V5 was expressed in these cells, the intracellular creatine concentration was significantly increased. Expression of AGAT without the V5 tag also trended towards increased creatine levels (~1.4-fold) but did not achieve statistical significance. These changes were much smaller than increases observed when AGAT and GAMT were both overexpressed. Thus, the low amount of endogenous GAMT expression can permit creatine synthesis, but flow through this pathway is dramatically improved by overexpressing both enzymes. These data indicate that GAMT levels limit the synthesis of creatine in this context. Considering these observations, we chose to co-express both AGAT and GAMT in primary fibroblasts to maximize creatine synthesis.

In both *SLC6A8* WT and mutant fibroblasts, overexpression of AGAT and GAMT markedly boosted intracellular creatine concentrations. For all experiments measuring this metabolite, we exchanged the media to creatine-free media 24 hours (HEK293T experiments) or 72 hours (fibroblast experiments) prior to measurement. The standard formulation of DMEM (as used here) does not contain creatine. However, FBS derives from a natural source and the concentration of small metabolites in FBS likely varies by lot [28]. As an estimate, the creatine concentration in adult bovine serum has been measured at approximately $200\,\mu M$ [29]. Thus, cells expressing a functional CT1 would have the ability to concentrate creatine from the media during propagation, prior to exchanging the media to creatine-free media. This would not be the case for the CTD patient fibroblasts we used for our experiments, as these cells are hemizygous for a complete loss-of-function allele for *SLC6A8*. The strong increase in intracellular creatine concentration we observed in these cells, and likely those observed in *SLC6A8* WT cells, were due to internal synthesis of creatine followed by retention of this metabolite intracellularly. Furthermore, the fold difference in creatine concentration measured in the media observed with AGAT and GAMT overexpression was lower in magnitude than the intracellular difference, indicating that diffusion from an extracellular compartment could not explain the intracellular increase. Rather, some diffusion out of the cytoplasm or background cell turnover/death is the likely source of higher media concentrations of creatine observed in cells overexpressing AGAT and GAMT. These data indicate that creatine can be synthesized and concentrated in cells lacking CT1.

## Limitations

We utilized immunofluorescence for AGAT and GAMT to localize these proteins in fibroblasts stably expressing our lentiviral constructs after blasticidin selection. Our immunofluorescence experiments were performed after several passages without

blasticidin selection, confirming that both transgenes were expressed without strong selection against expressing cells. However, we did not perform quantitative western blotting over time to study the relative stability of these transgenes and their equilibrium expression over time. Future studies could reveal whether AGAT and GAMT expression provides a positive or negative selective advantage for WT and CT1 mutant cells by measuring the level of expression, sampled throughout successive passages, after removing blasticidin selection. Of note, the measurements of creatine and total cellular protein we performed in fibroblasts were carried out after more than a month in culture without blasticidin. The marked increase in intracellular creatine we observed reflects persistent expression of these transgenes, but a more quantitative analysis would provide additional insight into the phenotypic consequences of AGAT and GAMT overexpression.

The kit we employed to measure creatine (both intracellular levels and those in the media) detects total levels of this metabolite. Intracellular creatine exists in a phosphorylated and unphosphorylated form. Due to this constraint, it is not known if the rise in intracellular creatine observed upon AGAT and GAMT overexpression results in the increase of only one or both metabolites and in what ratio. Given that both HEK293 cells and human fibroblasts express cytoplasmic creatine kinase (namely, CKB) [30,31], it would be expected that a portion of the synthesized creatine would exist as phosphocreatine. Future studies would be required to determine the relative abundance of creatine/phosphocreatine in cells overexpressing exogenous AGAT and GAMT.

Our prior studies indicated that certain cell types, such as oligodendrocytes, express both AGAT and GAMT at high levels *in vivo* [15]. Based on this observation, we hypothesized that overexpressing these enzymes would permit the internal synthesis of creatine, boosting its levels in cells with a genetic defect in *SLC6A8*. We have demonstrated that this is indeed the case for CTD patient derived fibroblasts, suggesting that this strategy could be used for gene therapy in CTD. Replacement of a functional copy of *SLC6A8*, however, would be a more direct therapeutic approach for these patients [11], if delivery and expression can recapitulate its endogenous pattern of expression. Although we did not observe any salient deleterious consequences in our cell culture experiments with AGAT and GAMT overexpression, this may not be the case *in vivo*, where precursor amino acids or related metabolic pathways may be more limited [32,33]. Indeed, if neurons are truly the primary end-user of creatine in the brain, why has expression of AGAT and GAMT been delegated to oligodendrocytes? It could be that the metabolic demands of creatine synthesis [33] are consequential for, or incompatible with, highly active cell types such as neurons. Future studies would need to provide a detailed analysis of the metabolic consequences of AGAT and GAMT in various cell types. We did not quantify the amount of intracellular arginine, glycine, methionine or SAM in AGAT and GAMT overexpressing cells. Indeed, these experiments are first steps towards a proof-of-principal along a longer continuum which would involve additional cell culture, organoid/tissue, *in vivo* pre-clinical and, ultimately, early phase clinical experiments. Although AGAT and GAMT overexpression may consume amino acid precursors, the degree to which this happens does not impair cell growth. A particularly important case is WT fibroblasts overexpressing AGAT and GAMT without precursor supplementation as this removes the potentially confounding effect of rescue from a state of creatine deficiency that could be occurring in *SLC6A8* mutant fibroblasts. Collectively our data indicate that AGAT and GAMT overexpression permits highly productive synthesis and retention of creatine in novel cell types, including those lacking CT1.

## Supporting information

**S1 Fig. Raw Data For** Fig 1. The figure includes (A) Raw uncropped western blots used to prepare Fig 1A. **(B)** Raw uncropped western blots used to prepare Figure 1B. The raw data for creatine and protein measurements is included. (ZIP)

**S2 Fig. Raw Data For** Fig 2. The figure includes (A) Raw uncropped western blots used to prepare Fig 2A. **(B)** Raw uncropped western blots used to prepare Figure 2B. The raw data for creatine and protein measurements is included. (ZIP)

**S3 Fig. Raw Data For** Fig 3. Raw tiff files containing the confocal Z-stack processed to produce the images in Fig 3.
(ZIP)

**S4 Fig. Raw Data For** Fig 4. The figure includes (A) Raw uncropped western blots used to prepare Fig 4A.**(B)** Raw uncropped western blots used to prepare Figure 4B. The raw data for creatine and protein measurements is included.
(ZIP)

**S5 Fig. Raw Data For** Fig 5. Raw tiff files containing the confocal Z-stack processed to produce the images in Fig 5.
(ZIP)

**S6 Fig. Raw Data For** Fig 6. Raw tiff files containing the confocal Z-stack processed to produce the images in Figure 6.
(ZIP)

**S7 Fig. Raw Data For** Fig 7. Excel sheet containing the raw confluence data recorded by Incucyte which was analyzed in Figure 7
(ZIP)

## Acknowledgments

We kindly thank members of the Rutter laboratory and the Baker laboratory for many helpful suggestions and discussion. We are sincerely grateful to Erica Swenson for editing the manuscript. CW, JS, and SAB were supported by a grant from the Association for Creatine Deficiencies. SAB was supported by a grant from the Uplifting Athletes' Foundation. SLJ, DA, and SAB were supported by funds from the University of Utah Department of Pathology. JR is an Investigator of the Howard Hughes Medical Institute.

## Author contributions

**Conceptualization:** Steven Andrew Baker, Chloe Wells, Jon Sorgenfrei, Jared Rutter.

**Data curation:** Chloe Wells, Jon Sorgenfrei.

**Formal analysis:** Steven Andrew Baker, Chloe Wells, Jon Sorgenfrei.

**Funding acquisition:** Steven Andrew Baker, Jared Rutter.

**Investigation:** Steven Andrew Baker, Chloe Wells, Jon Sorgenfrei, Sadie L. Johnson, Devin Albertson.

**Methodology:** Steven Andrew Baker, Jared Rutter.

**Supervision:** Steven Andrew Baker.

**Writing – original draft:** Steven Andrew Baker, Chloe Wells, Jon Sorgenfrei.

**Writing – review & editing:** Steven Andrew Baker, Chloe Wells, Jon Sorgenfrei, Jared Rutter.

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
