## [Decision Letter · Decision Letter 0]

7 Aug 2024

PONE-D-24-17122Gene Delivery of AGAT and GAMT Boosts Creatine Levels in Creatine Transporter Deficiency Patient FibroblastsPLOS ONE

Dear Dr. Baker,

Thank you for submitting your manuscript to PLOS ONE. After careful consideration, we feel that it has merit but does not fully meet PLOS ONE’s publication criteria as it currently stands. Therefore, we invite you to submit a revised version of the manuscript that addresses the points raised during the review process.

I would like to sincerely apologise for the delay you have incurred with your submission. It has been exceptionally difficult to secure reviewers to evaluate your study. We have now received two completed reviews; the comments are available below. The reviewers have raised significant scientific concerns about the study that need to be addressed in a revision.

Please revise the manuscript to address all the reviewer's comments in a point-by-point response in order to ensure it is meeting the journal's publication criteria. Please note that the revised manuscript will need to undergo further review, we thus cannot at this point anticipate the outcome of the evaluation process.

We look forward to receiving your revised manuscript.

Kind regards,

Miquel Vall-llosera Camps

Senior Staff Editor

PLOS ONE

Journal Requirements:

"CW, JS, and SAB were supported by a grant from the Association for Creatine Deficiencies. SAB was supported by a grant from the Uplifting Athletes’ Foundation.  SLJ, DA, and SAB were supported by funds from the University of Utah Department of Pathology. JR is an Investigator of the Howard Hughes Medical Institute."

"We kindly thank members of the Rutter laboratory and the Baker laboratory for many helpful suggestions and discussion. We are sincerely grateful to Erica Swenson for editing the manuscript. CW, JS, and SAB were supported by a grant from the Association for Creatine Deficiencies. SAB was supported by a grant from the Uplifting Athletes’ Foundation.  SLJ, DA, and SAB were supported by funds from the University of Utah Department of Pathology. JR is an Investigator of the Howard Hughes Medical Institute."

"CW, JS, and SAB were supported by a grant from the Association for Creatine Deficiencies. SAB was supported by a grant from the Uplifting Athletes’ Foundation.  SLJ, DA, and SAB were supported by funds from the University of Utah Department of Pathology. JR is an Investigator of the Howard Hughes Medical Institute."

"I have read the journal's policy and the authors of this manuscript have the following competing interests: JR is a founder of Vettore Biosciences and a member of its scientific advisory board. All other authors declare no competing interests."

5. We note that your Data Availability Statement is currently as follows: [All relevant data are within the manuscript and its Supporting Information files.]

Reviewers' comments:

Reviewer's Responses to Questions

**Comments to the Author**

1. Is the manuscript technically sound, and do the data support the conclusions?

Reviewer #1: Yes

Reviewer #2: Yes

2. Has the statistical analysis been performed appropriately and rigorously? 

Reviewer #1: Yes

Reviewer #2: Yes

3. Have the authors made all data underlying the findings in their manuscript fully available?

Reviewer #1: Yes

Reviewer #2: Yes

4. Is the manuscript presented in an intelligible fashion and written in standard English?

Reviewer #1: Yes

Reviewer #2: Yes

5. Review Comments to the Author

Reviewer #1: Reviewer comments

Title: Gene Delivery of AGAT and GAMT Boosts Creatine Levels in Creatine Transporter Deficiency Patient Fibroblasts

Journal: PLOS One

Manuscript number: PONE-D-24-17122

Manuscript type: Research Article

Authors: Wells et al.

Wells et al report increased creatine levels in HEK293T cells, wild-type fibroblasts and in fibroblasts of individuals with CRTR deficiency when co-delivered with GAMT and GATM. They report that this gene delivery strategy will overcome the need for CRTR.

There are few questions listed below:

1. The AGAT and GAMT are written italic, as this is a gene delivery, however the gene name for AGAT deficiency is GATM that needs to be corrected throughout the manuscript. If authors using AGAT as abbreviation of the enzyme, they can still use AGAT for enzyme/protein name.

2. Last sentence of the abstract is confusing if this is a gene therapy strategy for CRTR deficiency or for all CCDS as the last paragraph of introduction refers to CRTR deficiency not all CCDS. Please revise accordingly.

3. In material and methods, is there typo for patient numbers as control and 13-year old affected male abbreviated as P2. Please check and correct accordingly.

4. Even if those CCDS types 1, 2, 3 are used in OMIM, they are not used in the medical literature. Please change the terms used CCDS 1, 2, 3 to actual enzyme and transporter deficiencies throughout the manuscript, as those are not used consistently in the manuscript. Refer to GAMT and AGAT deficiencies as well as creatine transporter deficiency and remove all CCDS types throughout the manuscript.

5. Despite abstract reports that the study findings establish proof-of-concept for gene therapy for CRTR deficiency, discussion does not give any information how the overexpression of GAMT and GATM would be feasible gene therapy approach in CRTR deficiency as they will still need precursor therapy with arginine and glycine. Their intakes in each neuronal cell in the brain is not feasible as HEK293T and fibroblasts does not correspondence to brain neuronal cells. These experiments should have been performed in brain organoids to be able to show if creatine would increase in brain neuronal cells with the overexpression of GAMT and GATM.

6. Discussion repeats study results, which should be revised.

7. Why did not authors use iPSC derived oligodendrocytes for their experiments, if their method would allow creatine increases if the overexpression of GAMT and GATM would have any toxic effects on those cells?

Reviewer #2: This study intends to provide some proof of principle that expression of enzymes in the creatine synthesis pathway could be used in a gene therapy of the creatine deficiency syndrome. The genes for the two enzymes AGAT and GAMT are delivered to HEK293T cells or primary human fibroblasts from patients. The data provide evidence for AGAT and GAMT protein expression, their correct cellular localization, and their effect on a strongly increased intracellular creatine content.

While the study demonstrates very well that expression of AGAT and GAMT can dramatically boost intracellular creatine levels, it remains unclear whether this is not detrimental for the intracellular levels of the precursors of creatine synthesis. This is particularly important for arginine and SAM which are essential for many other metabolic pathways. There are also some issues with the immunofluorescence images and the creatine determination that should be addressed.

Major:

1. Introducing massive creatine synthesis in cells normally not synthesizing this metabolite may have drastic effects on the rewiring of metabolism. This important issue for such a study is not addressed in the experimental part. Somehow the authors are aware of the problem, since they supplement the creatine-free medium used with arginine, glycine and methionine. However, although "intracellular metabolites were extracted” (p.12) perfectly, creatine is the only molecule quantified. At least one of the critical precursors (arginine or SAM) should be quantified along with creatine in at least one of the experimental systems used.

Although the authors state that “boosting creatine synthesis does not have major deleterious effects on cellular physiology” (p.4), this is limited to the same cell number, same GAPDH abundance or same growth curve between control and transfected cells in standard medium. The authors admit that here may be problems in vivo “where precursor amino acids or related metabolic pathways may be more limited” and “It could be that the metabolic demands of creatine synthesis are … incompatible with, highly active cell types ...” (p.20). However, some of these potential important limitations could be already analyzed at the cellular level by looking at other metabolites.

2. Immunofluorescence results as described in the text are difficult to grasp from the figures.

- Figs 3, 5 and 6: In most panels labelled “DAPI”, there is no or only extremely faint staining visible, which precludes evaluation of the cell number and situation in the following panels

- Fig 3A,B: “Empty vector control cells displayed … cytoplasmic and low level nuclear signal was detected when these cells were stained for GAMT” (p.14) – This is not visible in the figures!

- Fig 3, 5A, 6A: “In cells stably expressing the AGAT and GAMT transgenes, tubular and punctate signal could be detected for AGAT throughout the cytoplasm” (p.14) and “In AGAT and GAMT transgenic CT1 mutant cells, the AGAT signal localized well to tubular/punctate structures consistent with mitochondrial targeting” (p.16) – Contrary to this text, there is very few to no staining of AGAT in Figs. 3A,B and 5A and 6A! Does this mean that many cells that express high levels of GAMT do not express AGAT (or only at very low level)?

3. Are the authors sure which sort of “intracellular creatine” they measure? Is this really only creatine, or rather a sort of “total creatine”, which is creatine + phosphocreatine? An overnight speedvac at room temperature risks to hydrolyze a large part of phosphocreatine, which is then detected as creatine. This issue should be addressed in the text.

Minor :

p.3: “GAA is then methylated on the same nitrogen” – better: “GAA is then methylated by GAMT on the same nitrogen”

p.4: “… if membrane channels that transport creatine are not expressed.” – membrane transfer of creatine uses a transporter, not a channel, so correct to “… if membrane transporters for creatine are not expressed.”

The immunofluorescence images are not very informative and could be limited to the 63x magnified parts of Figs. 3 and 5, once the above mentioned problems are solved.

6. PLOS authors have the option to publish the peer review history of their article (what does this mean? ). If published, this will include your full peer review and any attached files.

**Do you want your identity to be public for this peer review?** For information about this choice, including consent withdrawal, please see our Privacy Policy .

Reviewer #1: No

Reviewer #2: No

---

## [Author Response · Author response to Decision Letter 0]

12 Jan 2025

We thank the reviewers for their excellent and thoughtful comments. As we indicated to the editor when we submitted this manuscript, the funding for this project (and the research laboratory in which experiments were conducted) had ended. Therefore, we are not able to carry out additional experiments as requested. However, we have significantly revised the manuscript, incorporating the critiques of both reviewers. Our responses can be found below the reviewers’ comments which are copied here in blue.

Reviewer #1:

Wells et al report increased creatine levels in HEK293T cells, wild-type fibroblasts and in fibroblasts of individuals with CRTR deficiency when co-delivered with GAMT and GATM. They report that this gene delivery strategy will overcome the need for CRTR.

There are few questions listed below:

1. The AGAT and GAMT are written italic, as this is a gene delivery, however the gene name for AGAT deficiency is GATM that needs to be corrected throughout the manuscript. If authors using AGAT as abbreviation of the enzyme, they can still use AGAT for enzyme/protein name.

Thank you for this comment. You are completely correct. However, because of the similarity of the gene names for GATM and GAMT we and other authors (see PMID: 39402976, 31853708, and 26695944) often refer to the gene encoding AGAT by the same symbol with italics (namely, AGAT). To assist with clarity, we have revised the gene name to GATM throughout.

2. Last sentence of the abstract is confusing if this is a gene therapy strategy for CRTR deficiency or for all CCDS as the last paragraph of introduction refers to CRTR deficiency not all CCDS. Please revise accordingly.

We corrected the abstract to reflect the goal of the project and the introduction.

3. In material and methods, is there typo for patient numbers as control and 13-year old affected male abbreviated as P2. Please check and correct accordingly.

We apologize for being unclear here. We meant to indicate that the cells from this patient were obtained from Coriell at passage 2 (P2), not that this was patient 2. We have removed the abbreviation for clarity.

4. Even if those CCDS types 1, 2, 3 are used in OMIM, they are not used in the medical literature. Please change the terms used CCDS 1, 2, 3 to actual enzyme and transporter deficiencies throughout the manuscript, as those are not used consistently in the manuscript. Refer to GAMT and AGAT deficiencies as well as creatine transporter deficiency and remove all CCDS types throughout the manuscript.

We removed these references to CCDS type and replaced them by the suggested name. Thank you.

5. Despite abstract reports that the study findings establish proof-of-concept for gene therapy for CRTR deficiency, discussion does not give any information how the overexpression of GAMT and GATM would be feasible gene therapy approach in CRTR deficiency as they will still need precursor therapy with arginine and glycine. Their intakes in each neuronal cell in the brain is not feasible as HEK293T and fibroblasts does not correspondence to brain neuronal cells. These experiments should have been performed in brain organoids to be able to show if creatine would increase in brain neuronal cells with the overexpression of GAMT and GATM.

This is an excellent point, and we raised this concern in the original discussion. We are unfortunately unable to carry out additional experiments, but we have revised the wording in the manuscript to indicate that these are first steps towards a proof of principal along a lengthy continuum which would involve additional cell culture, organoid/tissue, in vivo pre-clinical and, ultimately, early phase clinical experiments. Please see the revised introduction and discussion sections.

6. Discussion repeats study results, which should be revised.

Thank you for bringing that to our attention. We have removed the portion of the discussion containing details from the results.

7. Why did not authors use iPSC derived oligodendrocytes for their experiments, if their method would allow creatine increases if the overexpression of GAMT and GATM would have any toxic effects on those cells?

A very thoughtful comment again. Oligodendrocytes naturally express high levels of Gatm and Gamt, and thus, these cells might be a natural experimental system to pilot exogenous overexpression. Our thought would be to overexpress these enzymes in iPSC derived neurons, which because of their metabolic demands, might be more susceptible to any detrimental effects of GATM and GAMT overexpression. Our laboratory did not have extensive experience with iPSC culture or differentiation, but this would be a reasonable experiment for future study.

Reviewer #2: 

This study intends to provide some proof of principle that expression of enzymes in the creatine synthesis pathway could be used in a gene therapy of the creatine deficiency syndrome. The genes for the two enzymes AGAT and GAMT are delivered to HEK293T cells or primary human fibroblasts from patients. The data provide evidence for AGAT and GAMT protein expression, their correct cellular localization, and their effect on a strongly increased intracellular creatine content.

While the study demonstrates very well that expression of AGAT and GAMT can dramatically boost intracellular creatine levels, it remains unclear whether this is not detrimental for the intracellular levels of the precursors of creatine synthesis. This is particularly important for arginine and SAM which are essential for many other metabolic pathways. There are also some issues with the immunofluorescence images and the creatine determination that should be addressed.

Thank you for your kind comments and excellent feedback.

Major:

1. Introducing massive creatine synthesis in cells normally not synthesizing this metabolite may have drastic effects on the rewiring of metabolism. This important issue for such a study is not addressed in the experimental part. Somehow the authors are aware of the problem, since they supplement the creatine-free medium used with arginine, glycine and methionine. However, although "intracellular metabolites were extracted” (p.12) perfectly, creatine is the only molecule quantified. At least one of the critical precursors (arginine or SAM) should be quantified along with creatine in at least one of the experimental systems used.

This is a highly valid point. If we were to continue this project, our laboratory would measure these precursors (especially arginine and SAM) in non-supplemented media to interrogate the consequences of AGAT and GAMT overexpression. Importantly, in Figures 7B, 7C, and 7E, we did not supplement the media beyond the original formulation of DMEM + 15% FBS. In Figures 7D and 7F we did supplement the media with arginine, glycine, and methionine. In all cases, the fibroblasts overexpressing AGAT + GAMT either grew equivalently (Figure 7B) or outgrew (Figure 7C-F) their empty vector controls. This suggests that although AGAT + GAMT overexpression may consume amino acid precursors, the degree to which this happens does not impair cell growth. A particularly important case is WT fibroblasts overexpressing AGAT + GAMT without precursor supplementation (Figure 7B) as this removes the potentially confounding effect of rescue from a state of creatine deficiency that could be occurring in SLC6A8 mutant fibroblasts.

We have not quantified these precursors in AGAT + GAMT overexpressing cells and have highlighted this important limitation in the discussion.

Although the authors state that “boosting creatine synthesis does not have major deleterious effects on cellular physiology” (p.4), this is limited to the same cell number, same GAPDH abundance or same growth curve between control and transfected cells in standard medium. The authors admit that here may be problems in vivo “where precursor amino acids or related metabolic pathways may be more limited” and “It could be that the metabolic demands of creatine synthesis are … incompatible with, highly active cell types ...” (p.20). However, some of these potential important limitations could be already analyzed at the cellular level by looking at other metabolites.

We completely agree. We included those comments throughout the manuscript to highlight these limitations. Our hope was to determine that overexpression of AGAT + GAMT could boost intracellular creatine at all (particularly in cells with loss-of-function of CT1). As we note above, a series of these experiments were carried out in fibroblasts with unsupplemented media. Still, even basal DMEM contains high concentrations of arginine·HCl (84 mg/L or 399 µM), glycine (30 mg/L or 400 µM), and methionine (30 mg/L or 201 µM) relative to plasma or CSF (9.2−54.7 µM, 2.3−72.7 µM, 1.3−14.2 µM for the latter, respectively per https://www.labcorp.com/resource/cerebrospinal-fluid-amino-acid-reference-intervals). Furthermore, the media above the cells provides an abundant volume from which these metabolites would need to be depleted. We are reassured that both reviewers appreciated these limitations from reading the manuscript as written. We hope that a reader of our work would be able to view our results in this context. Unfortunately, we have closed our laboratory and are unable to carry out additional experiments to explicitly address these concerns.

2. Immunofluorescence results as described in the text are difficult to grasp from the figures.

- Figs 3, 5 and 6: In most panels labelled “DAPI”, there is no or only extremely faint staining visible, which precludes evaluation of the cell number and situation in the following panels

Thank you for pointing this out. We found that our standard immunofluorescence protocol led to faint staining of the thin nuclei found in fibroblasts. We have adjusted the images to better highlight them by taking a Z-projection from a confocal stack.

- Fig 3A,B: “Empty vector control cells displayed … cytoplasmic and low level nuclear signal was detected when these cells were stained for GAMT” (p.14) – This is not visible in the figures!

We have adjusted the images to better demonstrate the low level of expression in control cells.

- Fig 3, 5A, 6A: “In cells stably expressing the AGAT and GAMT transgenes, tubular and punctate signal could be detected for AGAT throughout the cytoplasm” (p.14) and “In AGAT and GAMT transgenic CT1 mutant cells, the AGAT signal localized well to tubular/punctate structures consistent with mitochondrial targeting” (p.16) – Contrary to this text, there is very few to no staining of AGAT in Figs. 3A,B and 5A and 6A! Does this mean that many cells that express high levels of GAMT do not express AGAT (or only at very low level)?

There is some variation in expression likely attributable by lentiviral delivery, but we appreciate detectable AGAT in most cells. The PDF generated for review may have obscured the visualization of the signal. The original TIF figure reveals AGAT expression more clearly. However, along with reprocessing the DAPI and GAMT staining as requested above, we have reprocessed the AGAT signals using the same Z-projection. We hope you would agree this better demonstrates its expression. Thank you for making these suggestions.

3. Are the authors sure which sort of “intracellular creatine” they measure? Is this really only creatine, or rather a sort of “total creatine”, which is creatine + phosphocreatine? An overnight speedvac at room temperature risks to hydrolyze a large part of phosphocreatine, which is then detected as creatine. This issue should be addressed in the text.

This is an excellent point. We did not distinguish between creatine and phosphocreatine. Indeed, the kit we used (Sigma Aldrich Cat. #MAK079) measures total creatine in both the phosphorylated and non-phosphorylated form and thus we cannot interpret which of these two distinct (but highly related) metabolites is contributing to the signal. We have now addressed this important point in the discussion.

Minor :

p.3: “GAA is then methylated on the same nitrogen” – better: “GAA is then methylated by GAMT on the same nitrogen”

Much clearer, thank you!

p.4: “… if membrane channels that transport creatine are not expressed.” – membrane transfer of creatine uses a transporter, not a channel, so correct to “… if membrane transporters for creatine are not expressed.”

Agreed, thanks again. Fixed.

The immunofluorescence images are not very informative and could be limited to the 63x magnified parts of Figs. 3 and 5, once the above mentioned problems are solved.

We removed the low magnification images. We agree this helps to demonstrate the expression features more clearly in addition to the suggestions incorporated above.

---

## [Decision Letter · Decision Letter 1]

31 Jan 2025

Gene Delivery of AGAT and GAMT Boosts Creatine Levels in Creatine Transporter Deficiency Patient Fibroblasts

PONE-D-24-17122R1

Dear Dr. Baker,

We’re pleased to inform you that your manuscript has been judged scientifically suitable for publication and will be formally accepted for publication once it meets all outstanding technical requirements.

Kind regards,

Sharon DeMorrow

Academic Editor

PLOS ONE

Additional Editor Comments (optional):

Congratulations

Reviewers' comments:

Reviewer's Responses to Questions

**Comments to the Author**

1. If the authors have adequately addressed your comments raised in a previous round of review and you feel that this manuscript is now acceptable for publication, you may indicate that here to bypass the “Comments to the Author” section, enter your conflict of interest statement in the “Confidential to Editor” section, and submit your "Accept" recommendation.

Reviewer #1: All comments have been addressed

2. Is the manuscript technically sound, and do the data support the conclusions?

Reviewer #1: Yes

3. Has the statistical analysis been performed appropriately and rigorously? 

Reviewer #1: I Don't Know

4. Have the authors made all data underlying the findings in their manuscript fully available?

Reviewer #1: Yes

5. Is the manuscript presented in an intelligible fashion and written in standard English?

Reviewer #1: Yes

6. Review Comments to the Author

Reviewer #1: Reviewer comments

Title: Gene Delivery of AGAT and GAMT Boosts Creatine Levels in Creatine Transporter Deficiency Patient Fibroblasts

Journal: PLOS One

Manuscript number: PONE-D-24-17122R1

Manuscript type: Research Article

Authors: Wells et al.

Authors responded to this reviewer’s questions and revised the manuscript accordingly. There are no other questions from this reviewer.

7. PLOS authors have the option to publish the peer review history of their article (what does this mean? ). If published, this will include your full peer review and any attached files.

**Do you want your identity to be public for this peer review?** For information about this choice, including consent withdrawal, please see our Privacy Policy .

Reviewer #1: No

---

## [Editor Report · Acceptance letter]

PONE-D-24-17122R1

PLOS ONE

Dear Dr. Baker,

I'm pleased to inform you that your manuscript has been deemed suitable for publication in PLOS ONE. Congratulations! Your manuscript is now being handed over to our production team.

Kind regards,

on behalf of

Dr. Sharon DeMorrow

Academic Editor

PLOS ONE